# Optomechanical damping as the origin of sideband asymmetry

**João D. P. Machado⋆ and Yaroslav M. Blanter**

Kavli Institute of Nanoscience, Delft University of Technology,
Lorentzweg 1, 2628 CJ Delft, The Netherlands

⋆ m2501@gmx.com

## Abstract

Sideband asymmetry in cavity optomechanics has been explained by particle creation and annihilation processes, which bestow an amplitude proportional to 'n+1' and 'n' excitations to each of the respective sidebands. We discuss the issues with this as well as other interpretations, such as quantum backaction and noise interference, and show that the asymmetry is due to the optomechanical damping caused by the probe and the cooling lasers instead.

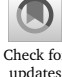

# 1   Introduction

Sideband asymmetry (SA) refers to the difference in spectral height of the side peaks accompanying a drive frequency. Whenever a resonator is driven coherently at a frequency $\omega_L$ and it is coupled to an oscillator (usually a mechanical resonator), the spectrum acquires peaks (the sidebands) at $\omega_L \pm \Omega$, with $\Omega$ the mechanical frequency. This phenomenon was first observed with trapped ions [1,2] and neutral atoms [3], where laser cooling unveiled motional sidebands around atomic transitions. With the emergence of optomechanics, SA was observed in systems with larger mechanical elements such as nanobeams [4,5], mechanical drums in LC-resonators [6], ultracold atoms [7], and membranes [8,9]. In the absence of any symmetry breaking mechanism, the sidebands' intensity would be expected to be equal. However, experimental observations reveal that one sideband is larger than the other. This imbalance was attributed to zero-point motion (ZPM) [4], and the reason for such quantum interpretation originates from stating that the measured spectrum is described by

$$S_{XX}(\omega) = \int_{\mathbb{R}} e^{i\omega t} \langle \hat{x}(t)\hat{x}(0) \rangle_{th} dt = \delta(\omega + \Omega)\bar{n}_{th} + \delta(\omega - \Omega)(\bar{n}_{th} + 1), \tag{1}$$

where $x$ is the displacement of the oscillator, and $\bar{n}_{th}$ its thermal occupancy. By identifying $\pm\Omega$ with the sidebands, ZPM would naturally explain SA, and consequently provide evidence for the quantum nature of the mechanical motion, regardless of its state.

The idea that ZPM could have an effect in the spectrum first came from the theory developed for single-ion spectroscopy [10], where the atomic transition decay rates associated with the sidebands differ from each other by an amount equal to ZPM. Nevertheless, the measurement of SA in these systems [1–3] employed linear detection techniques, whose measurement spectrum is not described by simple transition decay rates. The desire to observe quantum effects at the macroscopic scale lead to an increase in the efforts to cool mechanical resonators to the ground state, which in turn propelled developments in the optomechanical cooling theory [11,12]. The use of an asymmetric spectral function (such as Eq.(1)) in the cooling theory lead to the prediction that ZPM would create a spectral asymmetry. Within this weltanschauung, by cooling a resonator sufficiently, the rise of a spectral asymmetry testified the quantum nature of the resonator, as there could be no classical theory explaining this phenomenon [4], and it would make SA a paradigm for measuring temperature without requiring calibration [7–9].

The renewed attention paid to SA prompt alternative explanations, such as correlations between the resonator and the measurement probe [13]. This latter explanation, named *quantum backaction*, suggested a different cause for the asymmetry due to the use of a symmetric spectral density (thus different from Eq.(1)) and due to special commutation properties between the oscillator's position and the output noise. Nevertheless, the predictions of both explanations were identical, which does not allow to know the cause for SA. Building upon the interpretation of SA based on quantum correlations, the quantum nature of SA was disputed by the assertion that SA could have a purely classical origin, whose cause is the interference between electromagnetic and mechanical noises [14]. Thereafter, the quantum nature of SA and the role of ZPM was emphasised in this last interpretation, in attempt to reconcile it with the standard result [6,15]. The issues with the explanation of SA based on the interference between the noise channels are discussed in Sec.4. Unrelated to the previous alternatives, it was also advanced that laser phase noise can give rise to SA [15,16].

Regarding SA, a pervasive problem has plagued its interpretation: *a priori* definitions. The interpretation of SA differs for different operator orders (arising from different detector models [17]), and theoretical postulates on operator order do not necessarily match with the detection's physical mechanism.

In this article, we start by discussing the problems with the standard interpretation of SA, the connexion to how measurements are performed, and the spectral role of ZPM (Sec.2). We proceed to compute the response function for a system composed by two driven optical modes and a mechanical one in Sec.3. Considering the symmetric noise power spectral density for both cases, we show that SA naturally arises from the optomechanical damping caused by the interaction between the cavity and the mechanical oscillator, and that ZPM is not the cause of the asymmetry. Finally, we analyse in detail the issues of alternative interpretations, such as interference between noise channels and *quantum backaction*, in Sec.4. There, we also investigate the role of ZPM, and show that for a system composed by two modes (cavity + mechanical), but driven with multiple tones, the crosstalk between the tones leads to ZPM contributions to the spectrum significantly different from the ones predicted by Eq.(1).

Note that our analysis focuses only on the power spectral density obtained with linear detections methods. There are other methods closely related to SA which also make use of the term "sideband asymmetry", such as the difference between the photon count rates when an optomechanical system is driven with short pulses at the red and blue sidebands [18, 19]. These methods are outside the domain of the present work, because it is the instantaneous photon count rate that is measured instead of the frequency spectrum. As such, these methods do not suffer from the operator order issues discussed in Sec.2, and our conclusions do not apply to single photon detection because of the negligible backaction and noise associated with short pulsed weak probes.

## 2  The measurement problem

The problem over the nature of SA can be traced back to its measurement. In contrast to its classical counterpart, defining the power spectral density in a quantum framework poses a problem regarding the operators' order. Direct substitution of the fields by operators in classical formulas is an ambiguous procedure, as there are multiple possibilities and not all of them have a physical meaning. The problems with defining a quantum spectral density and the meaning of the distinct possibilities were raised before [14, 21], but remain unsolved. The usual way to address these issues is to take the specific measurement procedure into consideration. SA can be measured using well-known linear detection schemes such as homodyne or heterodyne detection. The quantum description of these techniques [22–24] typically focuses on the quadrature measurement and noise response on the time-domain, leaving issues with the frequency domain unmentioned. To measure the field quadrature $X(t) = a_s(t) + a_s^{\dagger}(t)$ (where $a_s$ is the annihilation operator for the signal), linear detection schemes combine the signal input with a local oscillator, and split the output into two detectors. The intensity difference between the detectors is proportional to $X(t)$, but it is in the frequency domain that the noise response is computed via the quadrature variance. This poses the problem of defining the variance of $X(\omega)$. As $X(\omega)$ is a non-Hermitian operator, there are different possible orderings, such as $\langle (X(\omega))^{\dagger} X(\omega) \rangle$ or $\langle X(\omega)(X(\omega))^{\dagger} \rangle$, as well as any convex linear combination. Each possibility produces a different outcome, but the uniqueness of the spectrum implies that only one should represent the observed spectrum. The noise power spectral density is obtained with the Fourier transform of the average of the product between different measurement outcomes. As $X(t)$ is Hermitian, the measurement outcome is a real number, and so is the product at different times. However $X(t)X(t')$ is not Hermitian, and therefore it can have non-real values as a possible outcome. Therefore $\langle X(t)X(0) \rangle$ cannot represent the physical measurement, and so can neither Eq.(1). The only Hermitian possibility that can represent the measurement is the symmetric combination of $X(t)X(t')$ with its Hermitian conjugate. Therefore, the most

suitable spectral density to describe the measurement is the symmetric spectral function

$$\bar{S}_{XX}(\omega) = \frac{1}{2}\Big\langle X(\omega)X(-\omega) + X(-\omega)X(\omega)\Big\rangle. \tag{2}$$

An alternative way to measure SA is with photodetection, and for this case the ordering issues are usually bypassed by choosing a detector model and establishing a link with the measurement outcomes. The typical detector model consists of a single qubit interacting briefly with the measured field via a weak dipolar coupling [25, 26]. The excitation probability $P_{exc}$ of a qubit in the ground state for short time-scales and coupled to a stationary random field is computed with perturbation theory, and it reads [21]

$$P_{exc} \propto \int_{-t}^{t} e^{i\epsilon t'}\langle X(t')X(0)\rangle dt'. \tag{3}$$

By identifying the qubit energy splitting $\epsilon$ with the frequency $\omega$, and $P_{exc}$ with the measured signal, Eq.(3) has been employed as a quantum spectral density. However, such toy model is unable to completely model the measurement because: (1) spectrometers are not composed of a single qubit, and a single qubit alone cannot provide the spectral density for a wide frequency range. Models with several qubits lead to higher order correlation functions [26] and higher spin states do not lead to Eq.(3) [27]; (2) Eq.(3) is valid for short time-scales, where the transition rate is a constant given by the Fermi golden rule. To obtain the spectral density, the system has to be monitored for extended time-intervals, after which the validity of this result breaks down; (3) other detection models lead to different operator orders, such as anti-normal order in photon counters [28].

Irrespectively of the model and definitions considered, ZPM should not be the cause of the asymmetry. Even though the measured field quadrature is associated with the operator $X$, the outcome of a measurement is a scalar $x$, and it is with the measurement record $x(t)$ that the spectrum is obtained. For the scalar $x(t)$, the order issue does not exist, the spectral density is well-defined, and there is no reason for ZPM to affect the sidebands differently for symmetrical frequency responses. Nevertheless, ZPM could affect the spectrum due to its role in the variance of $X$, and the exact effect of ZPM in the spectrum could be explained with the theory of quantum continuous measurements without the prescription of a spectral density. The formalism of continuous position measurements already exists [29, 30], as well as analogous formalisms to model photodetection [31]. However, we are unaware of similar approaches to describe the spectral measurements. A closely related approach to describe homo- and heterodyne detection featuring quantum trajectories is also available in the literature [32] but such approach still relies on operator order postulates to evaluate the spectrum and not solely on the measurement record.

## 3 Multimode measurement

To examine the nature of SA, we consider the optomechanical case where the sidebands are measured in transmission via a signal coming from an optical (or microwave) cavity coupled to a mechanical resonator. From input-output relations, the signal amplitude is proportional to the cavity field, and for linear couplings, the cavity field yields a linear relation with the mechanical displacement. For this reason, when the cavity is driven, the coupling to the mechanical resonator produces sidebands around the drive frequency that contain information about the mechanical motion. For the reasons exposed above, Eq.(2) shall be used to compute the spectrum.

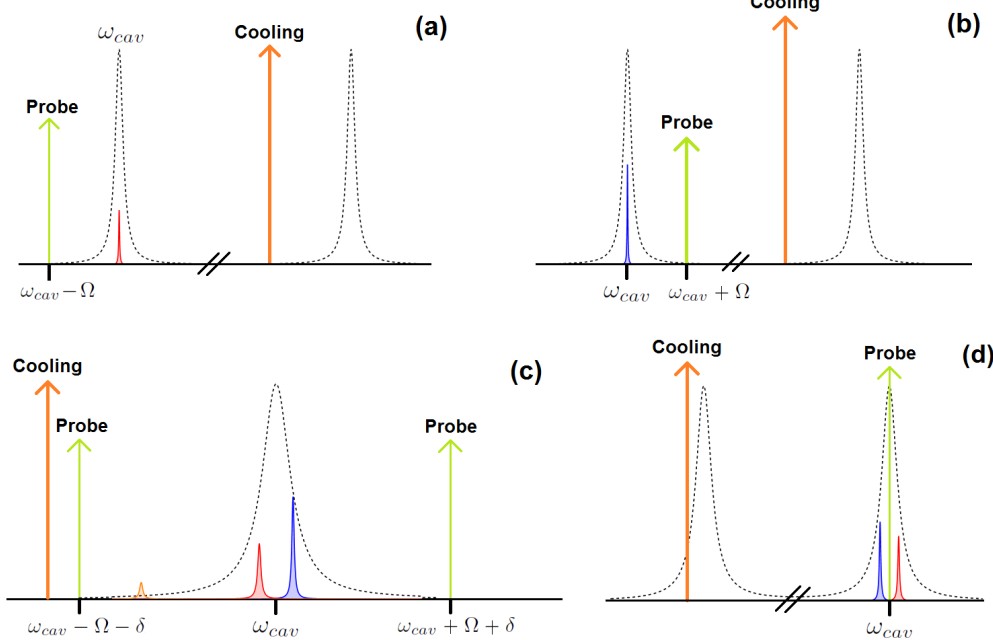

Figure 1: Different schemes to measure sideband asymmetry. The sidebands can be measured individually by placing the probe red(blue)-detuned (panel a (b)). Alternatively, a single cavity mode can be probed with two tones, which create sidebands within the cavity linewidth (panel c). The sidebands can also be measured directly with a probe tone on resonance (panel d). Cooling tones are represented for completeness.

A method to measure SA is to apply a probe beam at $\omega_{cav} \mp \Omega$ and measure the red(blue)-sideband at $\omega_{cav}$, and then tune the probe frequency to measure the other sideband (panels (a,b) of Fig.1). This way, each sideband is enhanced separately while the other sideband is off-resonant, and SA is measured more easily. Additionally, in a typical experimental situation, the mechanical resonator is cooled via an independent source. This source can be a distinct cavity mode driven at the red-sideband, and this case shall be analysed ahead. An alternative way to measure SA with only one cavity mode is to simultaneously drive the system with two probe tones (see Fig.1), with these two tones slightly detuned by $\delta$ from $\omega = \omega_{cav} \pm \Omega$ such that the sidebands do not overlap at $\omega_{cav}$. This method is analysed in Sec.4.

To better compare with the experimental situation, we consider two cavity modes: a cooling mode, and a read-out mode with a frequency far away from the cooling mode, such that no direct interaction between the modes occurs. The equations of motion describing this system are [6]

$$id_t b = \left(\Omega - i\frac{\Gamma}{2}\right)b - \sum_j g_j(a_j + a_j^\dagger) + \eta_b, \tag{4}$$

$$id_t a_j = \left(-\Delta_j - i\frac{\kappa_j}{2}\right)a_j - g_j(b + b^\dagger) + \eta_j, \tag{5}$$

where $\Gamma$ is the mechanical dissipation and $\Delta_j$, $\kappa_j$, and $g_j$ are the detuning, cavity linewidth and coupling strength for mode $j$. The detuning $\Delta_j = \omega_{L,j} - \omega_{cav,j}$ accounts for the shift of each mode $j$ from their respective drive reference frame, i.e. a reference frame with frequencies displaced from the drive frequencies $\omega_{L,j}$. Here and onwards, the cavity frequency shift produced by the static displacement of the resonator is included in $\omega_{cav,j}$. Furthermore, $b$ represents the phonon annihilation operator and $a_r, a_c$ the photon annihilation operators for

the read-out and cooling modes. At last, $\{\eta_j\}$ are the noise terms, with the properties

$$\langle \eta_j^\dagger(t)\eta_l(t')\rangle = \frac{\kappa_j}{2\pi}\bar{n}_j\delta_{jl}\delta(t-t'), \tag{6}$$

$$\langle \eta_j(t)\eta_l^\dagger(t')\rangle = \frac{\kappa_j}{2\pi}(\bar{n}_j+1)\delta_{jl}\delta(t-t'), \tag{7}$$

where $\bar{n}_j$ is the thermal occupancy for mode $j$. An analogous relation holds for the mechanical noise. Note that the system behaves linearly as long as the interaction is weak enough to prevent entering the amplification regime. When this regime is reached, an instability takes place (primarily at $\Delta = \Omega$), leading to a behaviour very different than just the creation of sidebands. Moreover, in the strong coupling regime, hybridisation between the cavity and the mechanics occurs, leading to additional spectral features, such as a frequency splitting at $\Delta = -\Omega$. As we are only concerned in addressing the SA issue, only the weak-coupling regime $g_j < \kappa_j, \Omega$ shall be considered, and since cooling occurs at the red-sideband, we set $\Delta_c = -\Omega$. Performing a Fourier transform in Eqs.(4-5) leads to the linear response function of the systems. The read-out field has the form

$$a_r(\omega) = q_1\eta_r(\omega) + q_2[\eta_r(-\omega)]^\dagger + q_3\eta_b(\omega) + q_4[\eta_b(-\omega)]^\dagger + q_5\eta_c(\omega) + q_6[\eta_c(-\omega)]^\dagger, \tag{8}$$

where the coefficients $q_j$ are displayed in App. A, along with the cavity response for the case where each sideband is driven separately.

In general, the read-out field does not have the same intensity at the red- and blue-sidebands because of the backaction from the cooling and read-out modes. This can be verified by evaluating, for example, the case with $\Delta_r = 0$ (corresponding to the experimental situation in [8,9]) in the limit $\Gamma \ll g_j, \kappa_j, \Omega$, which gives

$$\left|\frac{q_1(\Omega)}{q_1(-\Omega)}\right| = \left|\frac{A_- + iB_-}{A_+ - iB_+}\right| \neq 1, \tag{9}$$

where $A_\pm \approx 2(1+C_c)\Omega - \kappa_c(\kappa_r \pm \kappa_c C_r)/(4\Omega)$, $B_\pm \approx \kappa_r(1+C_c) + \kappa_c\left(\frac{1}{2}\pm C_r\right)$, and $C_j = 4g_j^2/(\Gamma\kappa_j)$ is the cooperativity for mode $j$. Thus, the asymmetry does not present a method for absolute self-calibrated thermometry.

When the sidebands are measured separately, $\Delta_r = \pm\Omega$. Using Eqs. (2) and (39-42), in the weak coupling regime and resolved sideband regime ($\kappa_j \ll \Omega$), the spectral density for each enhanced sideband is

$$\bar{S}_{XX}^\pm(\omega) \approx \kappa_o^2\kappa_r\left[\frac{(1+C_c)^2(\bar{n}_r+\frac{1}{2})}{(\omega\pm\Omega)^2+\frac{\kappa_r^2}{4}(1+C_c\mp C_r)^2} + \left(\frac{\Gamma}{\kappa_r}\right)^2\frac{C_r(\bar{n}_b+\frac{1}{2}+C_c(\bar{n}_c+\frac{1}{2}))}{(\omega\pm\Omega)^2+\frac{\Gamma^2}{4}(1+C_c\mp C_r)^2}\right], \tag{10}$$

where $X = a + a^\dagger$ and $\pm$ correspond to the blue(+) or red($-$) sidebands, and $\kappa_o$ is the cavity decay rate through the transmission port. As seen from Eq.(10), the only difference between the sidebands lies in the denominator, where the interaction modifies the linewidth of each sideband differently. It is also clear that ZPM does not contribute to the imbalance, and that the origin of the asymmetry for the weak-coupling and resolved sideband regime is the distinct effective optomechanical dampings for each sideband. Thus, instead of the height of the sidebands being given by $n$ and $n+1$, we find that the height of both sidebands is proportional to $n + \frac{1}{2}$, with $n$ the number of thermal excitations. A consequence of this difference is that at $T = 0$, the red sideband does not vanish. One could think that at $T = 0$, the absence of phonons would prevent the sideband to exist. However, as we are considering quadrature measurements instead of photon counting, the ZPM of the resonator affects the measurement of its position, and allows for the existence of the red sideband.



The asymmetry is quantified experimentally via the noise power $I^\pm$, which is obtained by integrating the area of the resonant sidebands $S^\pm$ over all frequencies. The asymmetry factor $\zeta$ quantifying the imbalance is given by

$$\zeta = \frac{I^+}{I^-} - 1 = \frac{2C_r}{1 - C_r + C_c}. \tag{11}$$

As optomechanical damping already occurs at the classical level [33], Eq.(11) shows a purely classical origin for SA. The standard result given by Eq.(1) predicts a temperature dependence $\zeta = 1/\bar{n}_b(T)$ for the asymmetry, and it is observed experimentally that SA becomes more prominent at low temperatures. The asymmetry imbalance in Eq.(11) has a temperature dependence, though indirect. For real physical systems, the bare mechanical quality factor $Q$ varies with the temperature, which makes $\zeta$ temperature dependent. Using Eq. (11) and considering the simplest case of a system probed resonantly ($C = 4g^2 Q/(\kappa\Omega)$) with a single read-out mode, in order to mimic this temperature behaviour, the temperature dependence of $Q$ must be

$$Q(T) \propto \frac{\kappa_r \Omega}{4g_r^2} \frac{1}{2\bar{n}_B(T) + 1} = \frac{\kappa_r \Omega}{4g_r^2} \tanh\left(\frac{\hbar\Omega}{2K_B T}\right). \tag{12}$$

The temperature dependence in Eq.(12) can be seen in Fig. 2 for $\Omega = 4$ GHz, $\kappa = 1$ MHz and $g = 10$ kHz, and it is qualitatively similar to the typical temperature behaviour of mechanical quality factors.

This temperature dependence of the mechanical damping rate has an important effect on the asymmetry. In [5], an increase by 3 orders of magnitude (from 1kHz to 1MHz) in the intrinsic mechanical damping with the increase in cavity temperature is reported, which drastically affects the asymmetry. For the parameter values mentioned above (similar to the ones reported in [5]), the asymmetry at 18K (where $\Gamma = 1$kHz) predicted by Eq.(11) is $\zeta \approx 1.3$, whereas at 210K ($\Gamma = 1$MHz), the asymmetry would drop to $\zeta \approx 0.001$. This change in the asymmetry is of the same order of magnitude as the typical experimentally determined values of $\zeta$, and it attests the impact of optomechanical damping. Note that the asymmetry reported in [5]

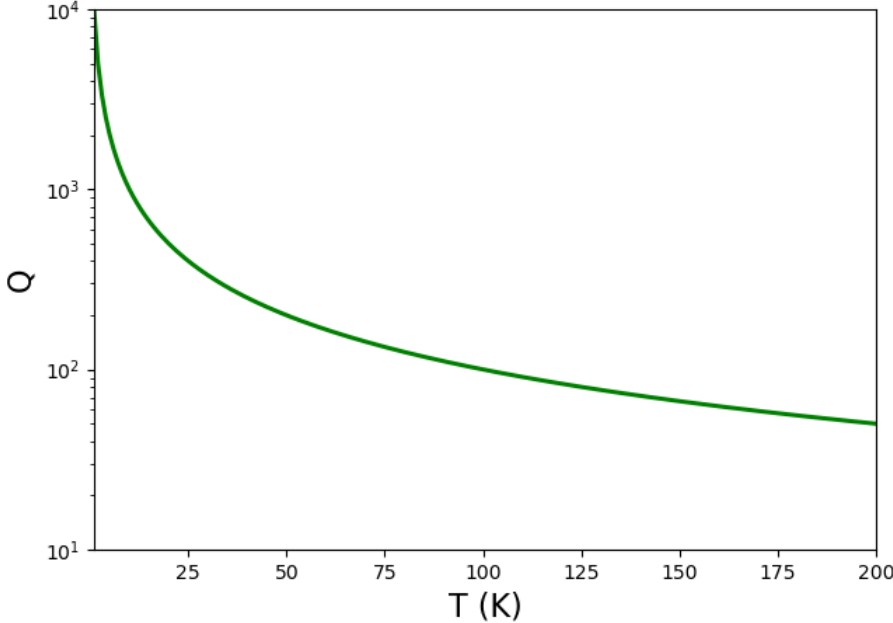

Figure 2: Dependence of the mechanical quality factor $Q$ with temperature for $\Omega = 4$GHz, $\kappa = 1$MHz and $g = 10$kHz, that exactly matches the standard result.

was not measured with the homodyne/heterodyne technique considered. However, quadrature measurements were employed in [4] to measure SA, the same type of optomechanical resonator was used, and similar values for the physical parameters were reported (the value of $g_r$ is not reported, only that $C_r \subset (0.01, 0.3)$). Therefore, optomechanical damping could explain the observed asymmetry.

We cannot say whether the temperature dependence of $Q$ fully explains all observed spectral asymmetries, but it plays a substantial role that should not be overlooked. Particularly, temperature calibrations must take this effect into account instead of erroneous ZPM effects.

# 4 Review of the quantum backation and noise interference interpretations

In this section, we review the explanation of SA based on noise interference and quantum backaction following the literature closely, and discuss the issues with these interpretations. We start with the analysis presented in [14], as the fully classical framework used gives the simplest clarifying start. Despite the new insight about the interference between the distinct noise channels and the possibility of SA without ZPM, the results and interpretation of [14] are not entirely correct. The physical situation is similar to the one in Sec. 3, but featuring a single optical mode $a$, which is used to measure SA in reflexion. For reflexion measurements, the measured read-out amplitude noise $H_r$ in the frequency domain is given by (cf. Eqs.(32,38) of [14])

$$H_r(\omega) = \kappa a(\omega) - \eta_a(\omega),\qquad(13)$$

where $\eta_a(\omega)$ is the optical noise as in the previous section. Here (as well as in [14]), we consider a single port cavity, where the input mirror is the only possible decay channel, the sidebands to be measured sequentially with the single driven mode (as in [14]), and disregard excess output noise. This is chosen to simplify the calculations and it has no effect overall. Keeping only the resonant terms, the equations of motion in the frequency domain at cavity/mechanical reference frame read (cf. Eqs.(30,31) of [14])

$$-i\omega a(\omega) = -\frac{\kappa}{2}a(\omega) + igb(\omega) + \eta_a(\omega),\qquad(14)$$

$$-i\omega b(\omega) = -\frac{\Gamma}{2}b(\omega) + iga(\omega) + \eta_b(\omega),\qquad(15)$$

for the red sideband, while for the blue sideband, the coupling terms are instead $ig[b(-\omega)]^\dagger$ and $ig[a(-\omega)]^\dagger$, respectively. Solving these equations, one finds the output field to be

$$H_\pm(\omega) = \chi_\pm(\omega)\eta_b(\omega) + \xi_\pm(\omega)\eta_a(\omega),\qquad(16)$$

with

$$\chi_\pm(\omega) = \frac{ig\kappa}{(-i\omega + \kappa/2)(-i\omega + \Gamma/2) \pm g^2},\qquad(17)$$

$$\xi_\pm(\omega) = \frac{(i\omega + \kappa/2)(-i\omega + \Gamma/2) \mp g^2}{(-i\omega + \kappa/2)(-i\omega + \Gamma/2) \pm g^2},\qquad(18)$$

where +(-) refers to the red(blue) sidebands, and $\omega$ is the deviation in frequency from the respective sideband. Computing the spectral function (c.f. Eq.(40) of [14]) with the output fields found above, leads to

$$S_\pm(\omega) = \Gamma|\chi_\pm(\omega)|^2 \bar{n}_b + \kappa|\xi_\pm(\omega)|^2 \bar{n}_a.\qquad(19)$$

This result differs from the one presented in Eqs.(41,42) of [14], where $S_{\pm}(\omega) = \kappa \bar{n}_a + \Gamma |\chi_{\pm}(\omega)|^2 (\bar{n}_b \pm \bar{n}_a)$ was expected. First, in Eqs.(41,42) of [14] the response susceptibilities presented for each noise channel are identical, while the different noise contributions to the spectral density have a different origin. The mechanical noise enters directly from the linear coupling between the read-out and the mechanical oscillator, and so the contribution to the read-out amplitude is $\xi$. The corrections to the optical noise come from the cavity modulation due to the mechanical resonator and add an extra term to the reflexion intensity, making the optical contribution to the read-out amplitude $1 + \tilde{\chi}$. As the noises are independent, the spectrum is given by the sum of their square amplitudes, and the effect of the interaction with the resonator is then $Re\{\tilde{\chi}\} + |\tilde{\chi}|^2$ for the optical noise, and $|\chi|^2$ for the mechanical noise. The only way for these contributions to interfere is if $Re\{\tilde{\chi}\} < 0$. However, $Re\{\tilde{\chi}\}$ is linear in $g$, while $|\chi|^2$ is quadratic in $g$, so the optical and mechanical contributions cannot be equal.

Second, explaining SA in terms of interference between different noise channels is misleading because the noise sources are uncorrelated, each of these has an independent contribution to the spectral function (the total spectral density is the sum of the modulus square of each noise contribution, and there are no cross terms).

Third, the cause of the sideband asymmetry is primarily optomechanical damping. This is visible from the $\pm g^2$ corrections to the linewidth in Eqs. (17,18), and it is even more clear from the peak height of each sideband

$$S_{\pm}(\omega = 0) = \kappa \left( \frac{4C}{(1 \pm C)^2} \bar{n}_b + \left( \frac{1 \mp C}{1 \pm C} \right)^2 \bar{n}_a \right), \tag{20}$$

with $C$ the cooperativity as defined above.

The reason for interpreting SA as resulting from interference between different noise channels is that the leading terms in the cooperativity in Eq.(20) can be written as $4\kappa C(\bar{n}_b \pm \bar{n}_a)$, which may give the impression of interference between optical and mechanical noise. However, this appearance of interference is only due to optomechanical damping. As mentioned above, mechanical noise enters the spectral density via the linear coupling between the read-out and the mechanical oscillator, which is linear in the cooperativity for small enough cooperativities, and identical for both sidebands. On the other hand, the response to optical noise comes primarily from optomechanical damping, which changes the linewidth of each respective sideband by $\pm C$. As these corrections to the sidebands' linewidth differ in sign, it appears as if there were an interference. However, this is not an entirely feasible explanation of SA. For optical modes, the thermal occupation at room temperature is $\bar{n}_a < 0.1$, which is much lower than any mechanical thermal occupation achieved so far, rendering the interference implausible. The true cause for SA is the optomechanical damping as discussed above, which is still present at $\bar{n}_a = 0$.

As the optomechanical interaction is linear in the regime considered, the noise response functions are identical for both the classical and quantum frameworks, with the only difference being the presence of ZPM, which amounts to the change $\bar{n}_i \rightarrow \bar{n}_i + 1/2$. However, there are still interesting features worth of discussion for the quantum case. For that, consider the physical situation of the experiment in [6], where a mechanical resonator is coupled to a LC resonator. The LC resonator is simultaneously driven with 3 tones: one tone to measure each sideband, and an additional tone to cool the resonator, as represented in panel (c) of Fig.1. For the sidebands to be enhanced, they must fall within the cavity linewidth, and in order to make them distinguishable, the tones must be slightly detuned from the red and blue sidebands by an amount $\delta$ (with $\Gamma \ll \delta \ll \kappa$ so that the peaks do not overlap and remain within the cavity linewidth). The cooling tone is also detuned from the sideband by $\delta_c$, with $\delta_c \gg \delta + \Gamma$, so the cooling tone does not overlap with the red-sideband probe tone. The analysis in [6] also accounts for loss through the different cavity ports, but that detail is irrelevant for the conclusions.

In this multi-tone case, the appearance of beats between the different tones $\varpi_j$ is inevitable, and the linear interaction of Eq.(5) can no longer be made time-independent. Thus, one must start with the full nonlinear interaction, and following [6], the equations of motion for the system are now

$$id_t a = \left(\omega_{cav} - i\frac{\kappa}{2}\right)a - g_0(b + b^\dagger)a + \eta_a(t) + \sum_j s_j e^{-i\varpi_j t}, \tag{21}$$

$$id_t b = \left(\Omega - i\frac{\Gamma}{2}\right)b - g_0 a^\dagger a + \eta_b(t), \tag{22}$$

where $\{s_j\}$ are the tones' amplitudes, and the sum in $j$ is over the $\varpi_j$ frequencies $\{\omega_{cav} \pm (\Omega + \delta), \omega_{cav} - \Omega - \delta_c\}$. The driving terms can be removed with the shift

$$a(t) = e^{-i\omega_{cav}t}A(t) + \sum_j e^{-i\varpi_j t}\alpha_j \quad, \quad b = \beta + e^{-i\Omega t}B(t), \tag{23}$$

where

$$\alpha_j = \frac{-s_j}{\omega_{cav} - \varpi_j - g_0(\beta + \beta^*) - i\frac{\kappa}{2}} \quad, \quad \beta = \frac{g_0}{\Omega - i\frac{\Gamma}{2}}\sum_j |\alpha_j|^2. \tag{24}$$

With this shift, part of the interaction is enhanced by $\alpha_j$ and linearised in $A$ and $B$. As the coupling $g_0$ is negligible in comparison to the other parameters, the nonlinear terms can be disregarded in favour of the enhanced linear terms. Additionally, because the system is driven by multiple tones, the spectrum will acquire higher sidebands due to the mixing between the different frequencies of the several tones. The amplitude of these higher harmonics is much lower than the first sidebands, and they are off-resonant, as their frequencies are far away from the cavity frequency. Therefore, all terms oscillating with a frequency equal to or higher than $\Omega$ are disregarded by invoking the rotating-wave approximation. With these approximations, the equations of motion in the frequency domain become

$$\left(i\omega - \frac{\kappa}{2}\right)A(\omega) = ig_0\left(\alpha_C B(\omega - \delta_C) + \alpha_R B(\omega - \delta) + \alpha_B[B(-\omega - \delta t)]^\dagger\right) - i\eta_a, \tag{25}$$

$$\left(i\omega - \frac{\Gamma}{2}\right)B(\omega) = ig_0\left(\alpha_C^* A(\omega + \delta_c) + \alpha_R^* A(\omega + \delta) + \alpha_B[A(-\omega - \delta)]^\dagger\right) - i\eta_b, \tag{26}$$

where the subscripts $R, B, C$ refer respectively to the red-sideband probe, blue-sideband probe, and cooling tone. As the phase of the driving amplitude will not play a role, we take $\alpha_j$ real and use the notation $g_j = g_0\alpha_j$ onwards. Further, the frequency dependence of the noise operators was omitted, as we only consider white noise. So far we have followed all approximations made in [6]. An additional approximation was made in [6], which is to consider that the different tones act independently, but we do not need (and make no use of) this approximation to compute the spectrum. Furthermore, we only evaluate the sidebands' height. Because the system cannot be made time-invariant, combinations between the different deviations of the sidebands appear. However, these higher order terms have an associated $\Gamma/\delta$ relative weight, and as the sidebands' visibility requires $\Gamma \ll \delta$, they can be disregarded. With this approximation (and taking $\delta \ll \kappa$), substituting Eq.26 in Eq.25 leads to

$$\frac{\kappa}{2}(1 + C_R)A(\delta) = i\eta_a + \frac{2g_R}{\Gamma}\eta_b - \frac{2g_R g_B}{\Gamma}(A(-\delta))^\dagger - \frac{2g_R g_C}{\Gamma}A(\delta_C), \tag{27}$$

$$\frac{\kappa}{2}(1 - C_B)A(-\delta) = i\eta_a + \frac{2g_B}{\Gamma}\eta_b^\dagger - \frac{2g_R g_B}{\Gamma}(A(\delta))^\dagger - \frac{2g_B g_C}{\Gamma}(A(\delta_C))^\dagger, \tag{28}$$

$$\frac{\kappa}{2}(1 + C_C)A(\delta_C) = i\eta_a + \frac{2g_C}{\Gamma}\eta_b - \frac{2g_C g_B}{\Gamma}(A(-\delta))^\dagger - \frac{2g_R g_C}{\Gamma}A(\delta). \tag{29}$$

Substituting the cooling tone amplitude $A(\delta_C)$, and computing the spectral amplitude measured in reflexion using Eq.(13), the sideband peaks in the limit of small cooperativities are

$$S(\delta) \approx \kappa \frac{4C_R(\bar{n}_b + \frac{1}{2}) + (1 - C_B - C_R + 2C_C - \sqrt{C_B C_R} - \sqrt{C_R C_C})^2 (\bar{n}_a + \frac{1}{2})}{(1 + C_R - C_B + 2C_C)^2}, \qquad (30)$$

$$S(-\delta) \approx \kappa \frac{4C_B(\bar{n}_b + \frac{1}{2}) + (1 + C_B + C_R + 2C_C - \sqrt{C_B C_R} - \sqrt{C_B C_C})^2 (\bar{n}_a + \frac{1}{2})}{(1 + C_R - C_B + 2C_C)^2}. \qquad (31)$$

The result above differs substantially from the spectral height found in [6] (cf. Eqs. (A21a, A21b)). A striking difference is the presence of terms such as $\sqrt{C_R C_B}$, which are due to a crosstalk between the different tones. These terms were neglected in [6] possibly due to the assumption of independence between the tones, but they have a substantial impact in the interpretation of SA. For simplicity, consider $C_C = 0$ and $C_R = C_B = C$. For this case, Eqs.(30,31) become $S(\pm\delta) \approx \kappa\left(4C(\bar{n}_b + \frac{1}{2}) + (1 - C \pm 2C)^2(\bar{n}_a + \frac{1}{2})\right)$, which makes the leading contributions in the cooperativity from ZPM to be $-\kappa C$ for the red-sideband and $3\kappa C/2$ for the blue sideband. In contrast, the contributions of ZPM of [6] for the spectral peaks are 0 for the red-sideband and $4\kappa C_B$ for the blue-sideband. The difference between the two results comes precisely from the $\sqrt{C_R C_B}$ term in Eqs.(30,31), which attests the significance of considering the crosstalk between the multiple tones. Even though higher-order sidebands were neglected, the probes tones act simultaneously and resonantly on the mechanical resonator, which in turn affects the read-out susceptibility. Thus, disregarding higher harmonics does not inhibit crosstalk between tones. The importance of this result is not merely a more accurate prediction of the spectrum; it has a more fundamental meaning. The impact of ZPM in the spectrum as computed in [6] predicts the same asymmetry as Eq.(1), which foments the idea of different yet equivalent interpretations of SA. However, the true spectrum displays an asymmetry different than the one predicted by Eq.(1), and thus the interpretations are not equivalent.

This difference in the sidebands' height is not restricted to the case of multitone measurements. It can be seen from Eq.(10) that the ZPM contribution to the sidebands' height is not identical to the one from Eq.(1), though they are equal in the leading terms in $C$. Nevertheless, taking only the leading terms in $C$ can be an inappropriate approximation. This is a critical issue in [6], where the small cooperativity approximation is not valid. The values reported in [6] ($\Gamma = 2\pi\times10$Hz, $\kappa = 2\pi\times860$kHz, $g_0 = 2\pi\times16$Hz and photon numbers $|\alpha_c|^2 = 4|\alpha_R|^2 = 4|\alpha_B|^2 = 4 \times 10^5$) lead to the cooperativities $C_C \approx 48$ and $C_R = C_B \approx 12$, which are far away from the small cooperativity regime $C < 1$. Further, as the sidebands are usually measured for low occupation numbers, $C \ll \bar{n}$ in order for the approximation to be consistent with the thermal calibration. At last note that, even though the peak height may match the standard result in some situations, the asymmetry imbalance (quantified in Eq.(11)) remains different and unaffected by ZPM.

## 5 Conclusion

The undisputed existence of SA is not a proof of a quantum nature of a physical system. By computing the symmetric noise power spectral density for a system of linearly coupled oscillators, we have shown that SA arises from the optomechanical damping caused by the laser drive, and that ZPM does not contribute to the asymmetry imbalance. For a linear system, the noise response function is identical in both classical and quantum descriptions, and as Stokes and anti-Stokes processes provide different amplitudes for the sidebands, an asymmetry naturally emerges. Additionally, we discuss how the asymmetry can have a temperature dependence

due to the natural temperature dependence of the mechanical decay rate. Though symmetric spectral densities have already been employed before, the asymmetry has been attributed to interference between the cavity noise and the mechanical resonator's noise [14] and to interference between the ZPM sources [6, 13, 34]. We have also discussed the problems with these interpretations, the role optomechanical damping plays in the interpretation, how the interaction and crosstalk between the cavity drive tones affect the asymmetry and the impact of ZPM.

Although our analysis is restricted to coupled harmonic oscillators, the same procedure can be extended to analyse the trapped ions and neutral atoms case [1–3]. The role of backaction has not yet been investigated in these systems, and a thorough analysis would clarify the nature of the asymmetry for this case. Note that for the case of trapped ions and atoms, their electronic quantum nature may lead to quantum signatures for the output light which are not an intrinsic feature of the light field (neither of the mechanical motion), much like in the case of Raman scattering [35].

## Acknowledgements

We thank C. Schäfermeier, A. Silva, G. Welker, J.P. Moura, and S. Sharma for the useful feedback provided.

### Funding information

This work was supported by the Dutch Science Foundation (NWO/FOM).

## A  Fourier coefficients and optical response function

The Fourier coefficients $\{q_j\}$ displayed at Eq.(8) are given by

$$q_1 = \beth(\omega)\Big[\Big(\Omega^2 - (\omega + i\tfrac{\kappa_c}{2})^2\Big)\Big(\big(\Omega^2 - (\omega + i\tfrac{\Gamma}{2})^2\big)(\Delta_r - \omega - i\tfrac{\kappa_r}{2}) + 2\Omega g_r^2\Big)$$
$$- 4g_c^2\Omega^2\Big(\Delta_r - \omega - i\tfrac{\kappa_r}{2}\Big)\Big], \tag{32}$$

$$q_2 = -\beth(\omega)2\Omega g_r^2\Big(\Omega^2 - \big(\omega + i\tfrac{\kappa_c}{2}\big)^2\Big), \tag{33}$$

$$q_3 = \beth(\omega)g_r\Big(\Delta_r - \omega - i\tfrac{\kappa_r}{2}\Big)\Big(\Omega + \omega + i\tfrac{\Gamma}{2}\Big)\Big(\Omega^2 - \big(\omega + i\tfrac{\kappa_c}{2}\big)^2\Big), \tag{34}$$

$$q_4 = \beth(\omega)g_r\Big(\Delta_r - \omega - i\tfrac{\kappa_r}{2}\Big)\Big(\Omega - \omega - i\tfrac{\Gamma}{2}\Big)\Big(\Omega^2 - \big(\omega + i\tfrac{\kappa_c}{2}\big)^2\Big), \tag{35}$$

$$q_5 = \beth(\omega)2\Omega g_r g_c\Big(\Delta_r - \omega - i\tfrac{\kappa_r}{2}\Big)\Big(\Omega + \omega + i\tfrac{\kappa_c}{2}\Big), \tag{36}$$

$$q_6 = \beth(\omega)2\Omega g_r g_c\Big(\Delta_r - \omega - i\tfrac{\kappa_r}{2}\Big)\Big(\Omega - \omega - i\tfrac{\kappa_c}{2}\Big), \tag{37}$$

where

$$\big(\beth(\omega)\big)^{-1} = \Big(\Omega^2 - \big(\omega + i\tfrac{\Gamma}{2}\big)^2\Big)\Big(\Delta_r^2 - \big(\omega + i\tfrac{\kappa_r}{2}\big)^2\Big)\Big(\Omega^2 - \big(\omega + i\tfrac{\kappa_c}{2}\big)^2\Big)$$
$$+ 4\Omega\Delta_r g_r^2\Big(\Omega^2 - \big(\omega + i\tfrac{\kappa_c}{2}\big)^2\Big) - 4\Omega^2 g_c^2\Big(\Delta_r^2 - \big(\omega + i\tfrac{\kappa_r}{2}\big)^2\Big). \tag{38}$$

With these coefficients, we can evaluate the case where each sideband is driven separately. At the enhanced red-sideband+cavity peak, the field amplitude for the read-out mode at $\Delta_r = -\Omega$ is

$$a_r(\omega) \approx Q_- \eta_r(\omega) - R_- \left( \eta_B(\omega) - \frac{2ig_c}{\kappa_c} \eta_c(\omega) \right), \tag{39}$$

while at the enhanced blue-sideband+cavity peak, the field amplitude of the probe at $\Delta_r = \Omega$ is

$$a_r(\omega) \approx Q_+ \eta_r(\omega) - R_+ \left( [\eta_B(-\omega)]^\dagger - \frac{2ig_c}{\kappa_c} [\eta_c(-\omega)]^\dagger \right), \tag{40}$$

where

$$Q_\pm \approx \frac{1 + C_c}{\omega \pm \Omega + i\frac{\kappa_r}{2}(1 + C_c \mp C_r)}, \tag{41}$$

$$R_\pm \approx \frac{2ig_r}{\kappa_r} \frac{1}{\omega \pm \Omega + i\frac{\Gamma}{2}(1 + C_c \mp C_r)}, \tag{42}$$

and considering the limit $g_j \ll \kappa_j \ll \Omega$ (weak coupling and resolved sideband regime).

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
