# Peer review of "Optomechanical damping as the origin of sideband asymmetry"

_SciPost Physics Core, doi:SciPost Phys. Core 5, 034 (2022)_

## Round 3 · Referee Report · Anonymous (Referee 3) · 2021-4-15

Strengths

This paper provides a new analysis of sideband asymmetry, something that has been of considerable to the quantum opto-mechanics community, and gives a new interpretation in which this effect is purely classical. The authors have a done an excellent job of comparing their analysis in detail with previous analyses so as to explain the reasons for the differences in interpretation. I think this new version is now appropriate for publication in ScPost Physics.

Weaknesses

none

Report

Acceptance criteria are met.

Requested changes

none

---

## Round 3 · Referee Report · Anonymous (Referee 1) · 2021-5-11

Strengths

Broadly speaking, I commend the authors on approaching a topical subject with renewed scepticism; putting forward a new interpretation that is entirely classical in nature and starkly different to the conventional understanding.

Weaknesses

A strong piece of evidence that would validate this model is a complete description of the experimentally observed scaling of SA with temperature (across multiple platforms like silicon-nitride membranes, superconducting circuits, photonics crystals, quantum liquids etc). At this point, I am totally unconvinced by the argument put forward by the authors that the observed SA is largely due to changes in mechanical dissipation.

The authors point a specific system as an example; PhC cavities (Ref. [5]). Indeed, PhC cavities have been used to observe SA, and they also exhibit extremely large changes in mechanical dissipation with temperature. However, PhC cavities are notoriously problematic in this regard. Other systems, such as silicon nitride membranes, do not have these problem (at least not the same extent) and they have also been used to measure SA with a temperature scaling consistent with conventional theory.

It's possible, I suppose, that the scaling with temperature could come from other factors too, such as changes in optomechanical coupling rate or thermal occupancy. However, to unambigously overturn conventional theory, a detailed description of all these effects would be critical. In this regard, the paper is lacking.

Report

As much as I would like to see a conventional theory completely overturned. I don't believe the authors have provided enough evidence to describe the plurality of experiments that have observed SA. So, unfortunately, I do not recommend publication in SciPost.
  • validity: low
  • significance: good
  • originality: good
  • clarity: good
  • formatting: good
  • grammar: good

Author:  João Machado  on 2021-05-24  [id 1461]

(in reply to Report 3 on 2021-05-11)

We must disagree with the referee regarding the evaluation of the validity of our results. The example given in the manuscript was chosen due to the particular difference in orders of magnitude of the mechanical damping variation with temperature, but that does not mean that our results are restricted to PhC cavities.

The referee mentions that other systems, in particular silicon nitride membranes, are not so affected by the temperature variation. We had already discussed an example concerning silicon nitride membranes before, which may have been overlooked by the referee. Even though the change in mechanical damping with temperature is much smaller for silicon membranes, it still has a significant effect. For clarity, we repeat the discussion here: In Appl. Phys. Lett. 107, 263501 (2015), a mechanical Q of 12.7e7 is reported at 14mK, and Q~4e7 at 200mK. This decrease by a factor of 3 in the mechanical quality factor is very little compared to the PhC example, but it still plays a role. The temperature range investigated is essentially the same as the one from the sideband measurements reported in ref.6, where the largest sideband ratio reported was of ~4.8, with the corresponding lowest value of ~1.4. According to our predictions (for the simplest case and disregarding interference and changes in thermal occupation), the estimated cooperativities would amount to ~0.66 at 20mK and ~0.16 at 200mK, which corresponds to an increase by a factor of ~4 for the intrinsic mechanical damping. This crude estimation of the variation of intrinsic mechanical damping with temperature is very much in line with the factor of ~3 decrease in the mechanical quality factor. Therefore, our theory can account for changes in sideband asymmetry in systems where the mechanical damping does vary drastically, such as silicon nitride membranes.

As the referee mentions, there might be indeed a temperature dependence of other physical parameters, such as the optomechanical coupling or the mechanical frequency, but that does not invalidate our theory. We have focused on the mechanical damping solely because it is the parameter with the largest variation with temperature and the most studied one. Thermal occupancy changes the height of the sidebands, but in most cases it does so symmetrically and thus does not affect the sideband asymmetry. The only exception we found was when interference occurs. We could indeed explore more sources of asymmetry, but we do not see how this will bring more clarity to the discussions. Furthermore, the discussion for other different systems has been made before and we were unable to find any contradictions between experimental observations and our claims.

At last, the referee displays a binary view concerning the validation of a theory by experiment. The referee insists that unless the prevailing theory is overthrown completely, our findings cannot be deemed valid. We wish to remind the referee that we put forward a fully consistent theoretical explanation that is NOT contradicted by experimental results. If the referee believes otherwise, we kindly ask the referee to point to a concrete experiment where such discrepancy is clear. Thus, we think it is erroneous to deem an alternative interpretation as invalid when it does not display any inconsistency nor it is contradicted by experiment.

---

## Round 3 · Author Response

Dear Editor,

We have carefully analised the comments from the referees and implement the necessary changes to answer to the questions posed. We hope that this new version clarifies the previous questions raised.

---

## Round 3 · List of Changes

Title - Optomechanical damping as the origin of sideband asymmetry

The title was changed to state up front what is the origin of sideband asymmetry instead of focusing on what is not.

Abstract

  • We included that we discuss "other interpretations, such as quantum backaction and noise interference" in the abstract.

Introduction

  • Expanded on the historical developments regarding the nature of sideband asymmetry and its connexion with zero-point motion right after Eq.(1), as well as a more detailed discussion of the previous work on sideband asymmetry.

  • "Thereafter, the role of ZPM was once again emphasised in the interference between the noise channels, in attempt to reconcile the explanation with the standard result" -> "Thereafter, the quantum nature of SA and the role of ZPM was emphasised in this last interpretation, in attempt to reconcile it with the standard result"

  • Moved the discussion on the photon-counting experiments from the conclusion to the introduction.

Section 2

  • General cleaning (mostly rephrasing and breaking sentences for clarity)

  • "Nevertheless, ZPM plays a role in the variance of $X$, and there is a link between $X(t)$ and the measurement outcome. As $X$ is monitored in time, a definite proof might rest in the theory of quantum continuous measurements." -> " Nevertheless, ZPM could affect the spectrum due to its role in the variance of $X$, and the exact effect of ZPM in the spectrum could be explained with the theory of quantum continuous measurements without the prescription of a spectral density"

Section 3

  • "A method to measure SA is to apply a probe beam at $\omega_cav-\Omega$ and measure the red-sideband at $\omega_{cav}$, and then tune the probe frequency to $\omega_{cav}+\Omega$ to measure the blue-sideband at $\omega_{cav}$" -> "A method to measure SA is to apply a probe beam at $\omega_{cav}\mp\Omega$ and measure the red (blue)-sideband at $\omega_{cav}$, and then tune the probe frequency to measure the other sideband"

  • The equation for the spectrum was rearranged for increased clarity

  • Included a discussion (at the end of the section) on the temperature dependence the quality factor needs to have in order to match the standard result, along with a figure displaying the temperature behaviour, and a comparison with typical experimental values.

  • "The standard result given by Eq.(1) predicts a temperature dependence $\zeta=1/\bar{n}_b(T)$ for the asymmetry, and it is observed experimentally that SA becomes more prominent at low temperatures. The asymmetry imbalance in Eq.(11) has a temperature dependence, though indirect. For real physical systems, the bare mechanical quality factor $Q$ varies with the temperature, which makes $\zeta$ temperature dependent. Using Eq. (11) and considering the simplest case of a system probed resonantly ($C= 4g^2 Q /(\kappa \Omega)$) with a single read-out mode, in order to mimic this temperature behaviour, the temperature dependence of $Q$ must be"

Section 4

  • Review of the quantum backaction and noise interference interpretations - Previous appendices B and C were merged and rewritten to present a thorough and detailed new section added addressing the issues with the quantum backaction and noise interpretations.

Conclusion

  • Adapted the conclusion to include the discussion in the section 4.

Overall - the term "backaction" referring to the source of the asymmetry was replaced by optomechanical damping for better clarity and to avoid confusion with the term "backaction" used to refer to interference between the ZPM of the cavity and the ZPM of the mechanics, as in (Phys. Rev. A86,033840 (2012); Phys. Rev. X4, 041003 (2014)).

---

## Round 4 · List of Changes

page 1:
Removed 1st sentence in the introduction
page 2:
"exegesis originates from proclaiming ex cathedra" -->" interpretation originates from stating"
"consequently prove" -- >" consequently provide evidence for"
removed "With the increase of experimental observations of SA, its quantum nature was deemed true."
"This explanation of SA based on the interference between the noise channels is incorrect due to inconsistencies and invalid approximations, which we discuss in Sec.4." --> "The issues with the explanation of SA based on the
interference between the noise channels are discussed in Sec.4."
page 3:
"and to assert that by definition, measurements follow a particular operator order, does not coerce a detector to measure that specific order. " --> "and theoretical postulates on operator order do not necessarily match with the detection's physical
mechanism."
removed "Theoretical interpretations should be based on the physical situation, instead of the experimental validation being subdued to theoretical postulates. This constitutes a problem for experimental validation, as biased premises have
been the starting point."
"classical formulas is dangerously arbitrary" --> "classical formulas is an ambiguous procedure"
page 9:
"First, Eqs.(41,42) of 14 cannot be correct because the response susceptibilities presented for each noise channel are identical," --> "First, in Eqs.(41,42) of 14 the response susceptibilities presented for each noise channel are identical,"
"it is erroneous to think in terms of interference between different noise channels" --> "explaining SA in terms of interference between different noise channels is misleading because"
"lead to the deceitful impression" --> "give the impression"
page 12:
"flaws" --> "problems"
Removed 1st sentence in the introduction
page 2:
"exegesis originates from proclaiming ex cathedra" -->" interpretation originates from stating"
"consequently prove" -- >" consequently provide evidence for"
removed "With the increase of experimental observations of SA, its quantum nature was deemed true."
"This explanation of SA based on the interference between the noise channels is incorrect due to inconsistencies and invalid approximations, which we discuss in Sec.4." --> "The issues with the explanation of SA based on the
interference between the noise channels are discussed in Sec.4."
page 3:
"and to assert that by definition, measurements follow a particular operator order, does not coerce a detector to measure that specific order. " --> "and theoretical postulates on operator order do not necessarily match with the detection's physical
mechanism."
removed "Theoretical interpretations should be based on the physical situation, instead of the experimental validation being subdued to theoretical postulates. This constitutes a problem for experimental validation, as biased premises have
been the starting point."
"classical formulas is dangerously arbitrary" --> "classical formulas is an ambiguous procedure"
page 9:
"First, Eqs.(41,42) of 14 cannot be correct because the response susceptibilities presented for each noise channel are identical," --> "First, in Eqs.(41,42) of 14 the response susceptibilities presented for each noise channel are identical,"
"it is erroneous to think in terms of interference between different noise channels" --> "explaining SA in terms of interference between different noise channels is misleading because"
"lead to the deceitful impression" --> "give the impression"
page 12:
"flaws" --> "problems"

---

## Editorial Decision

published